# The Association of Sugar-Sweetened Beverages to Children’s Weights Status Is Moderated by Frequency of Adding Sugars and Sleep Hours

**DOI:** 10.3390/children9071088

**Published:** 2022-07-20

**Authors:** Emmanuella Magriplis, Aikaterini Kanellopoulou, Venetia Notara, George Antonogeorgos, Andrea Paola Rojas-Gil, Ekaterina N Kornilaki, Areti Lagiou, Antonis Zampelas, Demosthenes B Panagiotakos

**Affiliations:** 1Laboratory of Dietetics and Quality of Life, Department of Food Science and Human Nutrition, Agricultural University of Athens, 11855 Athens, Greece; azampelas@aua.gr; 2Department of Nutrition and Dietetics, School of Health Science and Education, Harokopio University, 17671 Athens, Greece; katerkane@hua.gr (A.K.); gantonogeorgos@gmail.com (G.A.); dbpanag@hua.gr (D.B.P.); 3Laboratory of Hygiene and Epidemiology, Department of Public and Community Health, School of Public Health, University of West Attica, 12243 Athens, Greece; venotara@yahoo.gr (V.N.); alagiou@uniwa.gr (A.L.); 4Department of Nursing, Faculty of Health Sciences, University of Peloponnese, 22100 Tripoli, Greece; apaola71@yahoo.com; 5Department of Preschool Education, School of Education, University of Crete, 74100 Rethimno, Greece; ekornilaki@edc.uoc.gr; 6Faculty of Health, University of Canberra, Canberra, ACT 2617, Australia

**Keywords:** sugar-sweetened beverages, children, overweight, obesity, added sugars, sleep hours, additive effect

## Abstract

Adding extra sugars in milk and the moderating effect of sleep has yet to be investigated, setting the aim of this study. A total of 1361 school-aged children were included, aged 10–12 years old, by randomly sampling schools. Data were interview-obtained by trained personnel using a validated 17-item food frequency questionnaire, with specifics on type of milk and extra sugar additions. Analyses were stratified by average recommended hours of sleep. Predictive probability margins were obtained following necessary adjustments. Mean BMI was significantly lower, the level of SSB intake was higher, and the prevalence of adding sugars to milk four or more times per week was higher in children that slept ≥10 h. Most children (64%) consumed full fat milk, 21% had low fat, and 19.7% chocolate milk, with a significantly larger proportion of overweight or obese children consuming full fat or chocolate milk, also adding extra sugars four or more times per week (4.1% compared to 9.6%, and 12.2% compared to 39.5%, respectively). The predictive probability of being overweight or obese exponentially increased for children consuming >0.5 SSB/day while also adding sugars to their milk frequently, although this effect remained significant only for children sleeping <10 h/day. In conclusion, to accurately address the effect of SSBs on children’s body weight, frequency of any type of sugar addition in milk should be accounted for, as well as average sleep hours that may further moderate the effect.

## 1. Introduction

The obesity epidemic seen in children has been implicated with excessive consumption of free or added sugars [1,2]. This epidemic needs to be addressed since it has been reported that one in four obese children in many European countries are severely obese [3]. To understand the effect of sugar intake on obesity and compare findings from various studies, the differences between the two sugar expressions needs to be comprehended. Added sugars are those added during the processing of foods, or those packaged as such (table sugar, honey, corn syrup). Free sugars encompass added sugars and all naturally occurring mono- and di-saccharides present in foods and beverages, such as unsweetened fruit juice, but excludes sugars naturally present in intact fruits and in milk sugars (lactose). This is the updated definition by the World Health Organization (WHO) [4], also adopted by the United Kingdom Scientific Advisory Committee on Nutrition (UK SACN) [5].

Studies have reported that consumption of sugar-sweetened beverages (SSBs) [2], and other foods with added sugars [6,7], contributes to the increasing trend of overweight or obese children with associated health implications [8]. Overall excessive free sugar intake, mostly from beverages. has also been associated with obesity in adults [9] and adolescents [10]. Despite these findings, to date, the association between free and/or added sugar intake and obesity risk remains controversial in terms of type of food (solid versus liquid), type of sugar (naturally present or added), or both. A study investigating the type of sugar and its source among children in the US found that naturally occurring sugars from solids or liquids, including milk intake, did not affect obesity risk [6], whereas a study that included Australian adults reported that higher overall free sugar intake (added and naturally present) from beverages in relation to energy consumption was associated with higher Body Mass Index (BMI) [9]. A recent metanalysis in children and adults explicitly examined the association between soft drink intake with nutrition and health outcomes. They found that an increased consumption of soft drinks was associated with higher energy intake, increased body weight, and poor diet quality, the latter including lower milk and calcium intakes [1]. This is an area that requires further examination considering children’s high calcium requirements for growth, especially since the relationship between milk intake and anthropometric indices remains uncertain. Specifically, most studies do not differentiate between milk type and have not evaluated the addition of extra sugars [11], the latter being a potentially substantial contributor to added sugar intake, especially in children. A recent study that investigated the effect of frequently adding sugars to milk and to fruits [12] found a positive relationship with adiposity in school-aged children, although the potential additive effect of SSB intake was not examined.

In addition to the role of sugar intake on children being either overweight or obese, another key variable that may affect weight status by modifying appetite and potentially the urge for sweet tastes is inadequate sleep. In particular, inadequate sleep acts on the neuroendocrine system and increases appetite, unequivocally for fat [13] and potentially for simple carbohydrates [14]. Studies have shown that inadequate sleep in school-aged children is associated with an increased risk of being overweight or obese [15], and that adequate sleep ameliorates dietary quality [16] and decreases sweet and SSB intake [14,17], independent of other risk factors. Sleep patterns were evaluated by 41 surveys and found that children went to sleep at a later time as age increased, restricting sleep on school nights [18]. This decreasing trend in total sleep hours among older school-aged children may be further enhancing the problem of obesity.

Adding sugars to milk is a common practice among children and may overlap with SSB intake. The primary aim of the study is to therefore examine the additive effect of SSB intake and the frequency of adding sugars to milk on children’s weight status. Secondarily, to examine this effect in relation to sleep has not been examined, which thus formed the aim of this study.

## 2. Materials and Methods

School-aged children (10–12 years of age) were sampled by randomly selecting schools from a main urban area, Athens, and four semi-urban areas—Heraklion of Crete, Sparta, Kalamata, and Pyrgos—all from the Peloponnese geographical department. The list of schools for these areas was derived from the Greek Ministry of Education, with a total of 47 schools being selected using a stratified random sampling method, during school function between 2014 to 2016. Participation rate was above 95% in all schools that agreed to take part in the study. Overall study details, including data collection from parents, have been published in detail [18] but are not relevant in this study. Children were included upon obtaining a signed parental consent form. Information relevant to this study was available for 1361 children from the initial 1728 (79%) and is included in this study. A posteriori study power calculation was performed based on the statistical analysis performed and the available sample size (1361). A value of 86% was calculated to achieve an Odds Ratio (OR) of 1.2, with 0.3 probability of Ho, population standard deviation (σ) = 1, alpha set at 5%, and normally distributed variables.

### 2.1. Bioethics

The study was carried out in accordance with the Declaration of Helsinki and was approved by the Institute of Educational Policy of the Ministry of Education and Religious Affairs (code of approval F15/396/72005/C1).

### 2.2. Nutrition and Anthropometric Data

To acquire information on children’s eating habits, we used a validated semi-quantitative food frequency questionnaire (FFQ) which contained all foods and beverages that are more commonly consumed by the general child population. Responses were used to calculate the KIDMED score, a validated Mediterranean Diet Quality Index tool that approximates children’s nutritional diet in relation to the Mediterranean pattern [19]. This index was derived and validated in children and young adults aged 2–24 years of age residing in Spain and has been used by many Mediterranean countries. The index is based on a 16-item questionnaire that can be self-administered to decrease response error. Score summations ≥8 reflect optimal Mediterranean diet, whereas ≤4–7 indicates amendments are necessary and a score ≤3 reflects a low diet quality. Dietary information was collected from children with the help of trained interviewers and their educators. Information on SSBs was separately assessed and converted to level of weekly intake, considering a standard portion size (330 mL) for the purpose of this study. Children were also asked to report the type of milk consumed and were often helped by describing related packaging colors associated with fat percentage. Options included full fat regular milk, chocolate milk, and non- and low-fat milk (up to 2%) which were grouped together. Additionally, information on the frequency of adding sugar, honey, or chocolate powder to their milk, irrespective of type of milk consumed, was also asked and documented. These were grouped as added sugars (called milk additions in the text) and were categorized by frequency. High frequency was considered as ≥4 times per week; due to a lack of studies in this area, accounting recommendations by the WHO for limiting added sugar intake in children were used [4].

Each child’s weight (kg) was measured to the nearest 100 gr using a digital scale (Tanita), and height (cm) was measured to the nearest 0.1 cm using a portable stadiometer (Leicester Height Measure). All measurements, taken by trained investigators, were performed with skin-tight clothing and without shoes. Body Mass Index (BMI) was calculated as weight in kg divided by height in m^2^ to categorize their weight status as per the International Obesity Task Force (IOTF) tables [20].

### 2.3. Total Sleep and Other Variables

Children were asked to report the usual time they go to bed and the time they wake on weekends and weekdays separately. The recommended amount of sleep for 10–12-year-old school-aged children is between 9 and 11 h [21], and sleep compensation over weekends has been reported to partly ameliorate childhood obesity risk [22]. Based on this recommendation and the above findings, total sleep hours were derived as the average of total hours during the weekdays and during the weekend. Total sleep was primarily described in a continuous format and then categorized in a variable based on the median value of the recommendation that was developed (=10) to examine the potential modifying effect of children’s adherence to sleep hour recommendations on the study’s aim (the association of SSB intake and the frequency of milk additions on children’s weight status).

Age was calculated from date of birth and sex was documented as per the biological phenotype. Place of residence was also recorded. Children were asked to report an approximation of the total minutes they walked per day and were asked about these topics using closed-ended questions with values taken in 15 min intervals: 15–30 min, 30–45 min, 45–60 min, and ≥60 min per day. This was used to estimate daily activity and categorize them according to recommended daily movement for children. Specifically, children were categorized as those meeting the recommendation (≥60 min/day) compared to those reporting <60 min of daily movement. Maternal education was considered as a key socioeconomic factor that may affect association, as well as area of residence to account for lifestyle differences between urban and semi-urban areas. Educational level was categorized as those (i) having postsecondary or higher third level education and (ii) those with elementary and middle school education. More details may be found elsewhere [23].

### 2.4. Statistical Analysis

Continuous data were checked for normality using P–P and kernel density plots. Normally distributed data were presented as means with standard deviation and skewed variables as medians with their respective interquartile range (IQR). Student’s *t*-test and Mann–Whitney tests were used for between group comparisons, respectively. Categorical data were depicted as relative frequencies (n, %) and group differences were compared using the chi square test. Multiple logistic regression was used to examine the probability of being overweight or obese in children based on their consumption of sweets, SSBs, and the frequency of adding other sugars to their milk/drinks. Pearson correlation coefficient was used to assess the relationship between the children’s and parental adherence to the Mediterranean diet (KIDMED score vs. MedDietScore). A very good linear relationship was found and only the KIDMED score was therefore used in the logistic model to account for usual dietary habits, that may affect body weight, without parental habits so as to avoid collinearity. Interaction terms were used in the logistic regression analysis to examine the additive effect between regularly adding sugar, honey, and/or chocolate powder to milk and consuming SSBs. Potential confounding factors are either known a priori or as per the description table (included when between group differences were significant). The Bayesian Information Criterion (BIC) following the Likelihood Ratio Test (LRTest) was used to examine the significance of the model that included the interaction. The model containing the interactions had the lowest value of information criterion, and hence was considered better (it explained 42.7% or the variance compared to 33.4%). Predictive margin graphs (“marginsplot”) were developed to present the predicted probability of higher weight status by frequency of SSB intake and adding sugars to their milk with an exploratory graph that displays OR and 95% CI comparisons by sleep standard recommendations. All analyses were carried out using the STATA 14.0 statistical software package (StataCorp. College Station, TX, USA, LP) at a 5% significance level.

## 3. Results

Baseline characteristics of the children enrolled in the study, in total and stratified by the recommended hours of sleep (i.e., 10 h), are presented in Table 1. The mean hours that children slept was on average 9.7 (±1.0), with 56.3% sleeping less than 10 h per day. Sleeping <10 h per day was more frequent among boys, since 60.6% of the total males included slept less than 10 h compared to 52.7% of the total females (*p* = 0.004). Mean BMI was significantly lower in children that slept ≥10 h (18.9 kg/m^2^ compared to 19.5 kg/m^2^, *p* = 0.001), with 28.1% of the children being categorized as overweight or obese. Although no significant distributions were found between weight status and sleep category, a greater percentage of the overweight or obese children slept ≤10 h compared to healthy weight children (60.2% compared to 54.8%, respectively). The KIDMED score and the frequency of adding sugars to milk four or more times per week was higher among children who slept ≥10 h (*p* < 0.001 and 0.026, respectively) while weekly SSB intake was less (*p* = 0.004). Achieving more than 60 min per day of movement did not differ nor did maternal education level; however, more children residing in urban areas slept ≥10 h compared to those residing in semi-urban areas (58.8% compared to 51.5%, *p* = 0.01). No differences were found in daily milk servings (unsweetened) by sleep category.

In Figure 1, the type of milk children consumed in total and by weight status, as well as the percentage by weight status and milk type, for those that added extra sugars four or more times per week is presented. Of the total children included (n = 1361), 34 children (2.4%) reported not drinking milk at all, with no weight status differences found (2.5% healthy weight and 2.6% overweight or obese) and a total of 7.7% who added sugar to their milk four or more times per week (data not shown). Most of the children (64%) consumed full–fat milk, and although the majority were of healthy weight (67.2% compared to 57.5% being overweight or obese), a significantly higher proportion of overweight or obese children added sugars to their milk four or more times per week (4.1% healthy weight and 9.6% overweight or obese, *p* < 0.05). The proportion of children that consumed chocolate milk was 19.7% in total with no significant weight status differences (18.6% healthy weight and 22.6% normal weight or obese); however, 21% also added sugars to this milk four or more times per week, with a significantly higher proportion of overweight or obese children (12.2% healthy weight and 39.5% overweight or obese, *p* < 0.05). Lastly, 21% consumed low- or non-fat white milk, with the majority being overweight or obese children (25.7% vs. 19.3%, *p* = 0.009) of which 3.5% added sugars frequently as well. There were no weight status differences for this group (3.7% healthy weight and 3.1% overweight or obese).

In Figure 2, mean BMI levels are scattered by the level of daily SSB intake based on the category of total sleep hours. Fitted lines were drawn for each case for the frequency of adding extra sugars to the milk. The graph shows that children’s BMI when sleeping <10 h per day increased with a higher slope overall but was lower for those adding milk sugars less than four times per week. In comparison, although a linear trend was also seen for children who met average hour sleep recommendations, the slope was even less when milk additions were less than four times per week, a null almost negative trend resulted (meaning BMI was not associated with more frequent SSB intake when children slept 10–12 h per day and did not frequently add sugars to their milk (less than four times per week)).

All variables that significantly differed in the baseline table were considered in the multivariate logistic regression depicted in Table 2, where the odds of being overweight or obese compared to healthy weight by category of sleep hours was examined. A significantly lower likelihood of being overweight or obese was found among females (OR: 0.65; 95%CI: 0.51, 0.84) in total and for children who slept <10 h per day. Although SSB intake and the frequency of adding sugars to milk were not independently significant, a threefold additive effect was found for children adding sugars four or more times per week and consuming a higher level of SSBs per day (OR: 3.18; 95%CI: 1.71, 5.91). This remained significant and further increased to 3.49 (95%CI: 1.58, 7.71) for children sleeping <10 h per day. Lastly, children residing in semi-urban areas were 46% more likely to be overweight or obese compared to those in urban areas, with the likelihood ranging from 83% for those who slept <10 h to a non-significant 12% for children that slept ≥10 h per day on average.

In Figure 3a, the predicted probabilities of being overweight or obese are displayed in relation to daily level of SSB intake and frequency of using milk additions (extra sugars of any type) in milk (less than four times per week compared to four or more times per week). The probability statistically differs at the level of 0.5, meaning that consuming SSBs at a frequency level of ½ per day increases the probability of being overweight or obese significantly for children adding sugars to their milk four or more times per week compared to those that add sugars less than four times. The probability remains constant, at approximately 28% with 95% CI variations, with increasing SSB level for the group of children that less frequently adds sugars to their milk but drastically rises for those that add sugars four or more times per week, reaching 90% at a level of three SSBs/day. In Figure 3b,c, the same probability results are depicted in relation to meeting recommended sleep hours. Specifically, although the higher probability trend is shown for both groups, it remains significant only for children not meeting the recommended sleep hours (sleep < 10 h on average per day).

## 4. Discussion

The major findings of this study reveal that SSB intake alone, when consumed in moderation, does not increase the likelihood of being overweight or obese among school-aged children in the regions assessed; however, the effect is mediated by other sugars that may be frequently added to their milk. This effect is further moderated by average daily sleep hours, with children meeting the recommendations being further protected. More specifically, the probability of school-aged children becoming overweight or obese increases between 30% and 90% with increasing SSB daily intake when they also frequently add sugars to their milk, a beverage that children are recommended to consume to meet their calcium recommendations for growth. This is more prominent among children who sleep <10 h per day. All analyses accounted for area of residence, although lifestyle factors may significantly differ, as revealed in this study with a larger proportion of children that lived in the main metropolitan area meeting daily sleep hour recommendations.

To our knowledge, no other study has examined the association between child obesity and SSB intake whilst accounting for the frequency of adding sugars to their milk, although a review of epidemiological studies addressing short sleep duration has consistently reported higher energy consumption, a tendency for more palatable snacks, and an increased risk of obesity among other factors [13]. Another study also reported that frequently adding sugars to otherwise healthy food, such as fruit and milk, led to an increased likelihood of adiposity in school-aged children [12]. The study, however, did not evaluate the additive effect of SSB consumption or sleep in their analysis [12]. In accordance with our results, studies that have evaluated added sugar intake by type of food (solid vs. beverage) found that added sugars from (non-dairy) beverages were positively associated with children’s weight status [6,24]. Other studies, however, have reported an increased risk with consumption of any type of free sugars when consumed as a beverage [9]. Furthermore, the Healthy Lifestyle in Europe by Nutrition in Adolescence (HELENA) study failed to find an association between any type (solid or beverage) of free sugar intake and children’s weight status [10].

The neuroendocrine effect of sleep on appetite must be considered, in relation to the baseline characteristics of the children included in this study, when interpreting this study’s findings [13,14,25] since it has been shown that children that sleep less also tend to have more unhealthy dietary intakes [14,16]. In our study, the overall KIDMED score, meaning higher adherence to the Mediterranean diet, was moderate in both groups, with slightly higher scores achieved in children that slept adequately. However, a significantly greater proportion of children that slept less than the average recommended hours (<10) added sugars to their milk four or more times per week and consumed more SSBs daily, reaching up to four cans per day, compared to children that slept ≥10 h per day.

Most children enrolled in this study did not adhere to the dietary guidelines that recommend consumption of low–fat milk in comparison to findings by the National Health and Nutrition Examination Survey (NHANES) [26]. Specifically, NHANES reported that 65% of children consumed 1% or 2% milk compared to 21% in this study, 32% reported consuming whole milk compared to 64%, and 28% flavored milk compared to 19% in the present study. Milk additions were not included in this study. Furthermore, most of the children drinking chocolate milk, a milk type that provides approximately 25 gr of sugar per glass, also added extra sugars frequently: 1 child in 10 among those of healthy weight and 4 in 10 among those overweight or obese. The HELENA study, which aimed to describe the percentage of energy contribution from various fluids among European adolescents, found that 20.7% of the total energy provided by fluids was obtained from sweetened milk (including chocolate milk and flavored yogurt drinks) [27]. The authors, however, did not account for adding sugars to non-sugar containing milk types. It is also important to underline that a significantly lower percentage of children consuming non- or low-fat milk added other sugars to their milk, an area that requires more investigation. One can infer that this may be moderated by maternal educational level, a proposed significant risk factor for child obesity [28], although maternal education did not differ by group in this study.

A position paper by the European Society for Paediatric Gastroenterology, Hepatology and Nutrition Committee on Nutrition recommends that sugars should be consumed as fresh fruits instead of fruit juices or smoothies, as milk and unsweetened dairy products, and as part of a main meal [29]. The committee also recommended that sweetened milk products should be avoided, a recommendation supported by this study’s results. These changes are difficult to reinforce, since it has been shown that compliance to such recommendations is low [2] and the achievement of meeting the restrictive guidelines limiting added sugars to <5% of total energy intake has been questioned [30]. Overall, family-based obesity prevention interventions which target diet are most effective at younger ages (up to toddlers), as reported in a systematic review [31]. Among school-aged children, these family-based programs are not as effective [31]; therefore, other behavioral variables need to be addressed.

Study limitations need to be considered when interpreting these findings. These include the cross-sectional nature of the study which can be used to draw further hypothesis but no causal effects. Additionally, the specific amount of sugar used in milk was not assessed; therefore, data related to frequency was used. Lastly, since children reported their dietary intake, potential errors could have affected results, although this was minimal since each child’s intake was cross-referenced with parental intake and a high linear correlation was found. Prospective studies enrolling normal weight toddlers with multiple 24 h recalls, using the latest Automated Multipass Method along with food propensity questionnaires, are recommended.

## 5. Conclusions

In conclusion, type of sugar and type of food should be differentiated when addressing the effect of added sugars on weight status among children. Sleep patterns should not be overlooked in obesity research due to their multifactorial effect and should be included in lifestyle interventions [32]. Furthermore, to accurately address the effect of SSBs on children’s body weight, the frequency of any type of sugar addition to their milk should also be accounted for, as well as average sleep hours that may further moderate the effect. Considering that SSB intake, adding sugars, and total sleep hours are highly modifiable factors that overlap with one another, addressing total sleep hours (the simplest factor to change) may reduce the obesogenic effect of SSBs and adding sugars to milk. Community and family-based intervention trials, beyond diet and physical activity, are recommended to test this hypothesis.

## Figures and Tables

**Figure 1 children-09-01088-f001:**
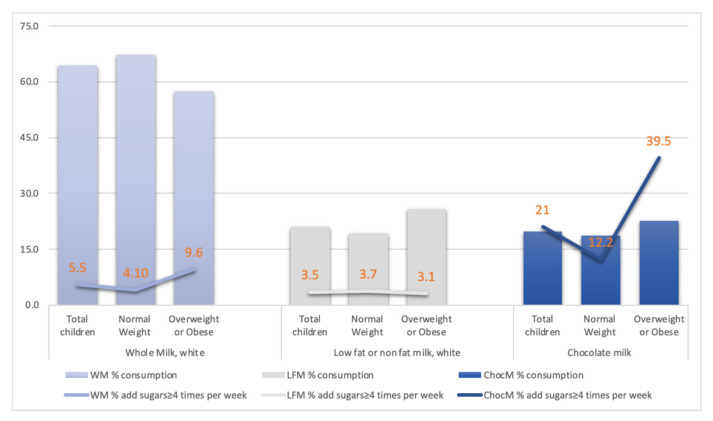
Proportion of children consuming whole, low fat, or chocolate milk in total and by weight status, with the proportion of children adding sugars four or more times per week in each case.

**Figure 2 children-09-01088-f002:**
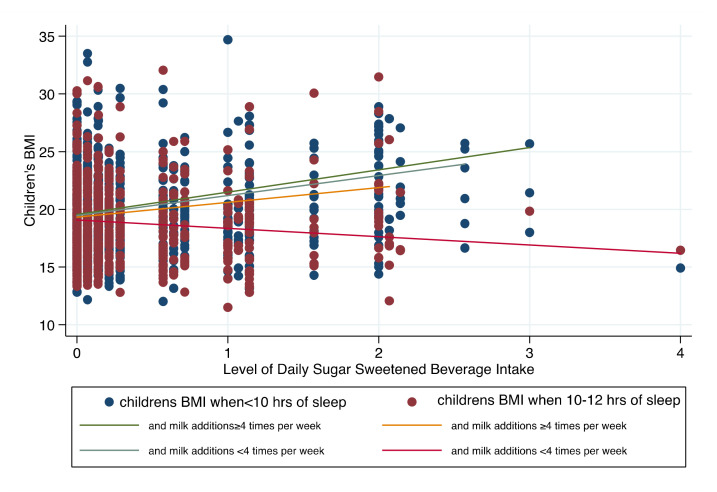
Scatter plot with fitted regression lines of mean BMI levels by the level of daily SSB intake based on the category of total sleep hours.

**Figure 3 children-09-01088-f003:**
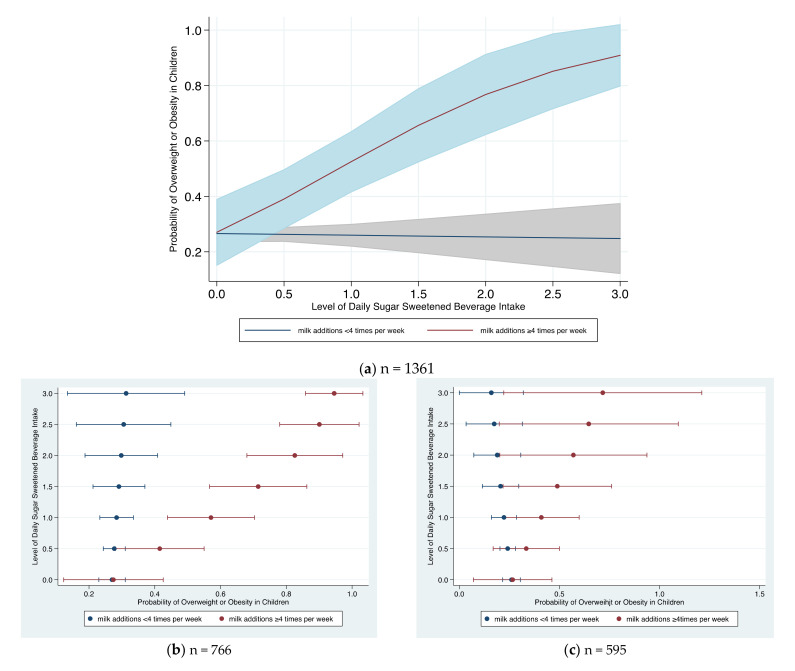
Predicted probability (mean and 95%CI) of children being overweight or obese with level of daily sugar-sweetened beverage intake by frequency of adding sugars to milk. Predictive probability of SSB with frequency of milk additions on being overweight or obese adjusted for sex, age, intake of sweets, KIDMED score, being active ≥60 min, residence, and average sleep hours. (**a**) Predictive probability of SSB with frequency of milk additions on being overweight or obese adjusted for sex, age, intake of sweets, KIDMED score, being active ≥60 min, residence, and average sleep hours. Complete model for (**b**) children sleeping <10 h/day and (**c**) children sleeping >10 h/day.

**Table 1 children-09-01088-t001:** Baseline characteristics of children enrolled in the study in total and stratified by recommended mean hours of sleep.

	Total Children n = 1361	Average Sleep Hours per Day ^1^ *Mean (sd): 9.7 (1.0)*		*p*-Value
		<10 Hours n = 766 (56.3%)	≥10 Hours n = 595 (43.7%)	
Age, mean (sd)	11.2 (0.8)	11.2 (0.8)	11.1 (0.8)	0.089
Sex, n (%)				0.004
Males, n%	612 (45.0)	371 (60.6)	241 (39.4)	
Females, n%	749 (55.0)	395 (52.7)	354 (47.3)	
BMI, kg/m^2^, mean (sd)	19.3 (3.4)	19.5 (3.5)	18.9 (3.3)	0.001
Weight status ^2^				0.068
Normal weight	979 (71.9)	536 (54.8)	443 (45.3)	
Overweight and obese	382 (28.1)	230 (60.2)	152 (39.8)	
KIDMED score, median (IQR)	4.0 (3.0, 6.0)	4.0 (3.0, 6.0)	5 (3.0, 6.0)	<0.001
Milk, unsweetened (servings per day)	1.0 (0.6, 2.0)	1.0 (0.6, 2.0)	1.0 (1.0, 2.0)	1.0
Sugar-Sweetened Beverage frequency per week, median (IQR)	1.0 (0.5, 4.0)	1.0 (0.5, 4.5)	1.0 (0, 2.0)	0.004
Milk additions, ≥4 times per week	105 (7.7)	70 (9.1)	35 (5.9)	0.026
Daily walking in min per day, >60 min	186 (13.7)	100 (53.8)	86 (46.2)	0.456
Postsecondary or higher third level education ^3^, n (%)	407 (45.5)	232 (45.3)	175 (45.8)	0.884
Children’s Residence				0.010
Urban	891 (65.5)	524 (58.8)	367 (41.2)	
Semi-urban	470 (34.5)	242 (51.5)	228 (43.7)	

^1^ Average sleep hours per week calculated as (mean sleep hours during weekdays + mean sleep hours during weekends)/2; ^2^ Weight status assessed based on IOTF standards; ^3^ Maternal educational level: mothers with at least a bachelor’s degree; Normally distributed data presented as mean (sd) with two group *t*-tests used to compare group differences; Skewed data presented as median (IQR) and Kolmogorov–Smirnoff test used to compare group differences; Categorical data presented with frequencies (n, %) and distribution differences tested with the chi square test. BMI: Body Mass Index; IQR: interquartile range; IOTF: International Obesity Task Force.

**Table 2 children-09-01088-t002:** Children’s odds of being overweight or obese while accounting for the additive effect of adding sugars to milk and SSB intake using a multivariable logistic regression model.

	Total Children		Children Who Sleep <10 Hours per Day		Children Who Sleep ≥10–12 Hours per Day	
Weight Status	Odds Ratio ^1^	[95% Confidence Interval]	Odds Ratio ^1^	[95% Confidence Interval]	Odds Ratio ^1^	[95% Confidence Interval]
SSB ^2^	0.97	0.75–1.25	1.08	0.78–1.49	0.81	0.53–1.26
Milk additions ≥4 times per week	1.02	0.54–1.93	1.02	0.45–2.33	1.03	0.36–2.91
Milk additions ≥4 times per week SSB ^2^	3.18	1.71–5.91	3.49	1.58–7.71	2.37	0.79–7.14
Sweets, frequency per week	0.83	0.60–1.16	0.89	0.59–1.35	0.67	0.37–1.22
Age, years	0.90	0.76–1.05	0.96	0.77–1.19	0.86	0.67–1.09
Sex, females vs. males	0.65	0.51–0.84	0.52	0.38–0.73	0.86	0.59–1.26
KIDMED score	0.96	0.90–1.01	0.96	0.89–1.04	0.95	0.87–1.04
Daily walking in min per day, >60 min	0.73	0.50–1.07	0.71	0.42–1.19	0.77	0.44–1.34
Residence ^3^	1.46	1.14–1.89	1.83	1.30–2.57	1.12	0.76–1.64
Sleep hours category	0.85	0.66–1.10				

Multivariate logistic model depicted in total. Bolded values depict those with significant effect on the model. ^1^ Sugar-sweetened beverages; ^2^ An interaction term between level of SSB intake per day and adding sugars to milk four or more times per week. Interaction term tested with Likelihood Ratio Test (LRT) and Bayesian Information Criterion (BIC). ^3^ Semi-urban vs. urban.

## Data Availability

Data can be made available upon request.

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
