# Peer review of "The Association of Sugar-Sweetened Beverages to Children’s Weights Status Is Moderated by Frequency of Adding Sugars and Sleep Hours"

_children, 2022, doi:10.3390/children9071088_

Round 1

Reviewer 1 Report

Overall, the authors are to be congratulated; this was a nice study with solid methodology, analysis and interpretation of results.

My main concerns:

1) Methods section should be more detailed with how things were done.  Some things are explained adequately, and others are not; the reader is told to go somewhere else (reference) to find the methodology. 

2) There were a lot of grammatical errors and awkward wording and sentence structure (like run-on sentences, etc.).  It would be extremely helpful to carefully revise manuscript so that it was more understandable and has better flow.

3) The tables and figures (figures especially) were a little hard to understand.  It would be helpful to have a little more description in the title an headings of tables and figures.  It seems like authors assume that readers know what tables and figures mean just because they present the data.  For example, in Table 1, on the 13th row, it says "high education level" with a footnote that explains.  It would be more clear if row 13 said "Mothers with College Degree". Also, on the 14th row, it would be more clear to say "Type of Residence of Child" vice "Residence". 

4) Last paragraph of discussion -- not sure what point authors are trying to convey.

Author Response

Reviewer 1

Overall, the authors are to be congratulated; this was a nice study with solid methodology, analysis and interpretation of results.

Authors: Thank you very much for your kind words and valuable Review. We have checked and have revised our manuscript considering all of Reviewers 1 comments and suggestions labeled in the pdf attached with your Review. All changes can be viewed with track changes in the newly submitted manuscript. Please note that tertiles for sugar addition in milk were not calculated, since the question was closed ended and frequency per week without amount was selected. Total milk servings per day have been included in Table 1.

My main concerns:

1) Methods section should be more detailed with how things were done.  Some things are explained adequately, and others are not; the reader is told to go somewhere else (reference) to find the methodology. 

Authors: We have provided some further details in methodology section, as recommended by the Reviewer. We now underline that we have provided details for all variables that are directly associated with this study hypothesis. Information for data regarding factors irrelevant to this specific study hypothesis have been previously published elsewhere, and of course referenced, now correctly (the references were affected as 19, although it was 18 (an extra number was inserted in reference 5 during the final formatting).

2) There were a lot of grammatical errors and awkward wording and sentence structure (like run-on sentences, etc.).  It would be extremely helpful to carefully revise manuscript so that it was more understandable and has better flow.

Authors: Apologies. We would like to thank the Reviewer for the constructive feedback. The manuscript has been edited grammatically, and all suggestions listed per page in the pdf were accounted for.

3) The tables and figures (figures especially) were a little hard to understand.  It would be helpful to have a little more description in the title an headings of tables and figures.  It seems like authors assume that readers know what tables and figures mean just because they present the data.  For example, in Table 1, on the 13th row, it says "high education level" with a footnote that explains.  It would be more clear if row 13 said "Mothers with College Degree". Also, on the 14th row, it would be more clear to say "Type of Residence of Child" vice "Residence". 

Authors: Thank you for the comment. We have included the Reviewer’s suggestions.

4) Last paragraph of discussion -- not sure what point authors are trying to convey.

Authors: We have revised the paragraph to clarify our suggestion which is that total sleep, is a key variable that needs to be assessed in child – obesity studies and prevention programs, since it may help decrease ssb intake and sugar additions to milk as well.

Reviewer 2 Report

This manuscript is titled "The association of Sugar Sweetened Beverages on Children’s 2 weights status is moderated by frequency of Adding Sugars 3 and Sleep Hours". My comments are outlined below:

- Authors need to clarify if Children provided verbal or written "Assent" to be included in the study.

- Authors must acknowledge and include the study limitations in the Discussion section.

- Authors must also list some future research directions to address these limitations.

- Minor English/grammatical errors exist that need to be proofed and corrected. For example: line 283 and 284 have the word "with increasing" duplicated.

Author Response

Reviewer 2

This manuscript is titled "The association of Sugar Sweetened Beverages on Children’s 2 weights status is moderated by frequency of Adding Sugars 3 and Sleep Hours". My comments are outlined below:

Authors: We would like to Thank Reviewer 2 for the time and effort spent to Review our manuscript.

- Authors need to clarify if Children provided verbal or written "Assent" to be included in the study. Authors: Parental consent was acquired prior to enrolling the child in the study (Lines 103-104).

- Authors must acknowledge and include the study limitations in the Discussion section.

Authors: Thank you for your comment. We have included the study limitations in Discussion (Lines 502-508).

- Authors must also list some future research directions to address these limitations.

Authors: Prospective study design has been recommended enrolling normal weight toddlers and assessing diet with multiple 24 hour recalls, along with FPQ’s, for greater dietary precision (Lines 508-510)

- Minor English/grammatical errors exist that need to be proofed and corrected. For example: line 283 and 284 have the word "with increasing" duplicated.

Authors: Apologies. We have carefully revised and edited the manuscript for grammatical errors. The duplicate has been removed.